# Predictions of Milk Fatty Acid Contents by Mid-Infrared Spectroscopy in Chinese Holstein Cows

**DOI:** 10.3390/molecules28020666

**Published:** 2023-01-09

**Authors:** Xiuxin Zhao, Yuetong Song, Yuanpei Zhang, Gaozhan Cai, Guanghui Xue, Yan Liu, Kewei Chen, Fan Zhang, Kun Wang, Miao Zhang, Yundong Gao, Dongxiao Sun, Xiao Wang, Jianbin Li

**Affiliations:** 1Institute of Animal Science and Veterinary Medicine, Shandong Academy of Agricultural Sciences, Jinan 250100, China; 2Shandong OX Livestock Breeding Co., Ltd., Jinan 250100, China; 3Department of Animal Genetics and Breeding, College of Animal Science and Technology, China Agricultural University, Beijing 100193, China; 4Yantai Institute, China Agricultural University, Yantai 264670, China

**Keywords:** prediction, milk, fatty acid content, mid-infrared spectroscopy, Chinese Holstein cow

## Abstract

Genetic improvement of milk fatty acid content traits in dairy cattle is of great significance. However, chromatography-based methods to measure milk fatty acid content have several disadvantages. Thus, quick and accurate predictions of various milk fatty acid contents based on the mid-infrared spectrum (MIRS) from dairy herd improvement (DHI) data are essential and meaningful to expand the amount of phenotypic data available. In this study, 24 kinds of milk fatty acid concentrations were measured from the milk samples of 336 Holstein cows in Shandong Province, China, using the gas chromatography (GC) technique, which simultaneously produced MIRS values for the prediction of fatty acids. After quantification by the GC technique, milk fatty acid contents expressed as g/100 g of milk (milk-basis) and g/100 g of fat (fat-basis) were processed by five spectral pre-processing algorithms: first-order derivative (DER1), second-order derivative (DER2), multiple scattering correction (MSC), standard normal transform (SNV), and Savitzky–Golsy convolution smoothing (SG), and four regression models: random forest regression (RFR), partial least square regression (PLSR), least absolute shrinkage and selection operator regression (LassoR), and ridge regression (RidgeR). Two ranges of wavebands (4000~400 cm^−1^ and 3017~2823 cm^−1^/1805~1734 cm^−1^) were also used in the above analysis. The prediction accuracy was evaluated using a 10-fold cross validation procedure, with the ratio of the training set and the test set as 3:1, where the determination coefficient (R^2^) and residual predictive deviation (RPD) were used for evaluations. The results showed that 17 out of 31 milk fatty acids were accurately predicted using MIRS, with RPD values higher than 2 and R^2^ values higher than 0.75. In addition, 16 out of 31 fatty acids were accurately predicted by RFR, indicating that the ensemble learning model potentially resulted in a higher prediction accuracy. Meanwhile, DER1, DER2 and SG pre-processing algorithms led to high prediction accuracy for most fatty acids. In summary, these results imply that the application of MIRS to predict the fatty acid contents of milk is feasible.

## 1. Introduction

Lipids in milk provide a major source of energy and the essential structural components for the cell membranes of the newborns in all mammalian species. They also confer distinctive properties to dairy foods that affect further processing procedures [1]. Milk fat is rich in many fatty acids that are important to human health [2,3,4]. Studies have shown that more than 400 different fatty acids have been identified in milk fat, but most of them only appeared in trace amounts [5], where around 12 kinds of fatty acids in bovine milk fat presented at above a 1% concentration [6]. Moreover, changes in milk fatty acids may also affect cow health and energy statuses [7].

Currently, several techniques have been developed to measure fatty acids in milk, including high performance liquid chromatography (HPLC), gas chromatography (GC), near-infrared spectroscopy (NIRS), mid-infrared spectrum (MIRS), etc. [8,9,10]. Chemical methods (e.g., HPLC and GC) provide high measurement accuracy for fatty acid contents of bovine milk, but their pretreatments are multifarious and costly, causing difficulties in realizing the high-throughput measurements [11,12]. Of note, infrared spectroscopy-based measurement methods show advantages of providing rapid and low-cost predictions of milk fatty acid contents [13]; thus, they have become the promising technologies for high-throughput measurements, but they still need to be optimized to improve their prediction accuracy.

The utilization of infrared spectroscopy to predict the milk fatty acid contents in dairy cattle has been reported in many studies. Coppa et al. (2010) established a prediction equation for milk fatty acid contents based on the NIRS from 468 milk samples that predicted the total milk fatty acids, SFA, MUFA, PUFA, C18:1, and conjugated linoleic acid (CLA), with R^2^ values greater than 0.88. Soyeurt et al. (2006) developed a fatty acid prediction model using 600 milk samples from 275 cows of 6 breeds to predict C10:0, C12:0, C14:0, C16:0, C16:1cis-9, C18:1, C18:2cis-9, SFA (saturated fatty acids), and MUFA (monounsaturated fatty acids), based on MIRS data, with the cross-validated coefficients of determination (R^2^) of 0.62 ~ 0.94. Subsequently, Soyeurt et al. (2011) investigated the MIRS prediction of fatty acids across various cattle breeds, production systems, and countries. They summarized that the usefulness of the built equations providing the best prediction accuracy for animal breeding and milk payment systems was R^2^ ≥ 0.75 and 0.95, respectively [4]. For the Dutch cattle breeds (Dutch Friesian, Meuse-Rhine-Yssel, Groningen White Headed, and Jersey), Maurice-Van Eijndhoven et al. (2020) updated the calibration equations from the European project RobustMilk [4] using the enlarged datasets and validated their usefulness to predict most milk fatty acids. De Marchi et al. (2011) used 267 milk samples from Brown Swiss cattle to predict fatty acids by MIRS and suggested the implementation of the used prediction models in milk recording schemes on fatty acid contents information for breeding purposes. Fleming et al. (2017) used MIRS to predict fatty acid contents from 373 cows of four breeds and obtained the cross-validation R^2^ of 0.60~0.90 for most individual fatty acid models. In addition, the genetic correlations among milk fatty acids predicted by MIRS were also explored in a large-scale milk sampling (*n* = 34,141) of New Zealand dairy cattle, where they implied the application of MIRS as the phenotypic proxy for the genetic selection of fatty acid contents [14]. In the Chinese Holstein population, Du et al. (2020) estimated the heritability of MIRS and several milk production traits, i.e., protein, fat, and lactose percentages, along with their genetic correlations. They found that MIRS heritability ranged from 0 to 0.11 and genetic correlations varied significantly [15]. In sheep, ewes, and goats, MIRS was also used to predict the fatty acid profiles for the establishment and validation of the predictive models [16,17,18].

Previous studies used a partial least square regression model (non-integrated learning model) [19,20] to investigate the effects of different spectral preprocessing methods on the prediction equation accuracy [4,5,21,22,23]. However, the combined effects of the regression models and spectral preprocessing methods on the prediction equation accuracy for different fatty acids has rarely been explored, especially for the milk fat of Chinese Holstein cows. Therefore, the objective of this study was to investigate the prediction methods under the optimal strategy to predict milk fatty acids with high accuracy based on the MIRS data from the dairy herd improvement (DHI) database of Chinese Holstein cattle and to potentially provide the high-throughput measurements of a large amount of milk fatty acid phenotypic data; thereby, our study enabled milk fatty acid traits to be feasibly recorded for genetic evaluations of such traits in dairy cattle breeding programs in China. To the best of our knowledge, this is the first time the MIRS predictions on fatty acids of two types of fatty acid measurements (g/100 g of milk and g/100 g fat) have been investigated with five pre-processed algorithms and two ranges of wavebands (4000~400 cm^−1^ and 3017~2823 cm^−1^/1805~1734 cm^−1^) using four regression models in Chinese Holstein cattle.

## 2. Results

### 2.1. Statistical Description

After quantification by the GC technique, statistical descriptions of individual and grouped fatty acid contents expressed as milk-basis (g/100 g of milk) and fat-basis (g/100 g of fat) are summarized in Table 1. The mean values of the individual fatty acid contents varied from 0.003 (C11:0, C20:1, C20:5n3, and C18:3n6) to 0.877 (C16:0) and their variation coefficients varied from 5.837% (C24:0) to 35.416% (C10:0), when they were expressed as milk-basis (g/100 g of milk). For grouped fatty acid contents, the mean values varied from 0.060 (SCFA) to 1.627 (SFA), and their variation coefficients ranged from 25.514% (PUFA) to 33.392% (SCFA) (Table 1). Similarly, the mean values of individual fatty acid contents varied from 0.094 (C20:5n3) to 28.620 (C16:0), and their variation coefficients varied from 13.802% (C16:0) to 44.207% (C22:1n9), when they were expressed as fat-basis (g/100 g of fat). For grouped fatty acid contents, the mean values varied from 1.934 (SCFA) to 52.710 (SFA), and their variation coefficients ranged from 12.978% (MCFA) to 19.365% (LCFA) (Table 1).

### 2.2. Predictions of Milk Fatty Acid Contents

The best prediction accuracy obtained by the optimal strategy from the test set for each fatty acid is summarized in Table 2, after considering different pre-processing algorithms, MIRS ranges, and regression models. In total, 16, 7, 6, and 2 fatty acids achieved the best prediction accuracy when the RFR, LassoR, PLSR, and RidgeR models were used, respectively. Similarly, the DER2, DER1, SG, SNV, and MSC algorithms resulted in 9, 8, 8, 4, and 2 fatty acids for best prediction accuracy, respectively. In addition, 22 fatty acids obtained the best prediction accuracy when they were expressed as g/100 g of milk (milk-basis), but only 9 fatty acids when expressed as g/100 g of fat (fat-basis). For most fatty acids (16/31), the ensemble learning model (RFR), with higher robustness and generalization, produced higher prediction accuracy than those predicted by the non-ensemble learning models (Table 2).

In Table 2, the best prediction accuracy (R^2^) for the optimal strategy showed R^2^ values from 0.62 (C20.3n6) to 0.91 (C20.5n3) in the test set for 28 fatty acids, where only 6 fatty acids showed R^2^ values higher than 0.8, including C12:0 (0.84), C20:0 (0.82), C22:0 (0.86), C20:5n3 (0.91), UFA (0.82), and LCFA (0.83). For R^2^ values higher than 0.75 and RPD values higher than 2, we found 17 fatty acids in total: C8:0 (R^2^ = 0.77 and RPD = 2.11), C10:0 (R^2^ = 0.77 and RPD = 2.07), C12:0 (R^2^ = 0.84 and RPD = 2.50), C14:0 (R^2^ = 0.78 and RPD = 2.05), C18:0 (R^2^ = 0.77 and RPD = 2.08), C20:0 (R^2^ = 0.82 and RPD = 2.35), C22:0 (R^2^ = 0.86 and RPD = 2.66), C24:0 (R^2^ = 0.80 and RPD = 2.20), C18:1n9c (R^2^ = 0.77 and RPD = 2.00), C20:1 (R^2^ = 0.76 and RPD = 2.04), C20:5n3 (R^2^ = 0.91 and RPD = 3.06), SFA (R^2^ = 0.76 and RPD = 2.01), UFA (R^2^ = 0.82 and RPD = 2.15), MUFA (R^2^ = 0.79 and RPD = 2.06), SCFA (R^2^ = 0.77 and RPD = 2.04), MCFA (R^2^ = 0.75 and RPD = 2.00), and LCFA (R^2^ = 0.83 and RPD = 2.29) (Table 2).

Table 3 shows the best prediction accuracy of different prediction models for each fatty acid, using training and test sets. All prediction accuracies (R^2^ and RPD) after four regression model analyses (RFR, PLSR, LassoR, and RidgeR), based on two types of fatty acid measurements (g/100 g of milk and g/100 g fat), two ranges of wavebands (4000~400 cm^−1^ and 3017~2823 cm^−1^/1805~1734 cm^−1^), and five spectral pre-processing algorithms (DER1, DER2, MSC, SNV, and SG), are listed in Appendix A. In the training set, the R^2^ values ranged from 0.18 to 0.79, with a mean of 0.58, and RPD values ranging from 1.08 to 2.18, with a mean of 1.59, when expressed as milk-basis (g/100 g of milk). Similarly, R^2^ values ranged from 0.13 to 0.90, with a mean of 0.47, and RPD values from 1.07 to 3.20, with mean of 1.52, when expressed as fat-basis (g/100 g of fat). In the test set, R^2^ values ranged from 0.14 to 0.84 with mean of 0.66 and RPD values from 1.04 to 2.50 with mean of 1.78 when expressed as milk basis (g/100 g of milk). Similarly, the R^2^ values ranged from 0.15 to 0.91 with mean of 0.49 and RPD values from 1.07 to 3.06 with mean of 1.52 when expressed as fat basis (g/100 g of fat) (Table 3). Additionally, the MIRS and processed MIRS after DER1, DER2, and SG pre-processing algorithms are shown in Figure 1.

## 3. Materials and Methods

### 3.1. Milk Samples and Fatty Acids

Milk samples were collected from 336 Holstein cows on a farm in Shandong Province, China, including one small tube (30 mL) and one large tube (50 mL) from each cow. After sampling, all tubes were immediately stored in liquid nitrogen (−196 °C) and delivered to our experimental lab for further analysis within 6 h. In this study, to maintain analysis consistency, none of the 672 collected milk samples received any preservative additions, and the milk samples in the 30 mL and 50 mL tubes were used to measure fatty acid contents and MIRS, respectively.

A total of 24 kinds of fatty acid contents, which included C8:0, C10:0, C11:0, C12:0, C13:0, C14:0, C14:1, C15:0, C16:0, C16:1, C17:0, C18:0, C18:1n9c, C18:2n6c, C20:0, C18:3n6, C18:3n3, C20:1, C22:0, C20:3n6, C20:4n6, C22:1n9, C20:5n3, and C24:0, were measured and quantified in each milk sample from the 30 mL tubes using the GC technique. Due to the limitations of GC technique, the apparent missing values were replaced by the averaged values of the whole fatty acids that had been quantified by the GC technique. The outliers of quality control for the fatty acids were defined by the mean reference values ± two standard deviations. For each milk sample from the 50 mL tubes, 899 raw data points for MIRS values in the complete waveband range of 4000~400 cm^−1^ were obtained by Bentley spectrometers (Bentley Instruments Inc., Chaska, MN, United States), following the routine methodology (e.g., 30 min preheating and sufficient shaking before operation). Afterwards, additional raw MIRS values, as the measurement replicates, were also obtained using the same milk samples. Finally, two raw MIRS values were transformed by the Fourier method [24] for further pre-processing steps.

Here, the GC methodology for the quantification of fatty acid contents in our study was similar to those in other studies [4,25]. The outputs of the GC technique were generated by analyzing the methyl esters from the fat in the milk following ISO Standard 15884 (ISO–IDF (International Organization for Standardization–International Dairy Federation), 2002). Normally, the GC technique is used as the gold standard for fatty acid measurements because of its high accuracy, even for low contents [26,27], while the MIRS method is more rapid and less expensive [13,21].

According to the saturation conditions of hydrocarbon chains, fatty acids are classified as saturated fatty acids (SFAs), unsaturated fatty acids (UFAs), monounsaturated fatty acids (MUFAs), and polyunsaturated fatty acids (PUFAs) [5]. According to the carbon chain lengths, fatty acids are classified as short chain (4 to 10 carbons) fatty acids (SCFAs), medium chain (11 to 16 carbons) fatty acids (MCFAs), and long chain (more than 16 carbons) fatty acids (LCFAs). Consequently, 7 fatty acid groups for 24 kinds of the above fatty acids were obtained (Table 4).

### 3.2. Predictions of Milk Fatty Acid Contents Using MIRS Data

Each fatty acid content quantified by the GC technique was converted from g/100 g of milk (milk-basis) to g/100 g of fat (fat-basis) using the fat contents determined by MIRS. The final MIRS values (the averaged values of two transformed MIRS replicates using the same milk sample) were processed using five spectral pre-processing algorithms, i.e., first-order derivative (DER1), second-order derivative (DER2), multiple scattering correction (MSC), standard normal transform (SNV), and Savitzky–Golsy convolution smoothing (SG). In order to compare the influence of each pre-processing algorithm, we used them individually to process the final MIRS values. Two types of fatty acid measurements (g/100 g of milk and g/100 g fat), with the five pre-processed spectra above and two ranges of wavebands (4000~400 cm^−1^ and 3017~2823 cm^−1^/1805~1734 cm^−1^), were analyzed using four regression models, i.e., random forest regression (RFR), partial least square regression (PLSR), least absolute shrinkage and selection operator regression (LassoR), and ridge regression (RidgeR). The determination coefficient (R^2^) and residual predictive deviation (RPD) were used to evaluate the metrics of the four regression models. Prediction accuracy was assessed using a 10-fold cross validation procedure with the ratio of the training set and the test set as 3:1. The GC quantification technique, fatty acid measurements, spectral pre-processing algorithms, fatty acid prediction methods, and prediction accuracy assessments are summarized in Figure 2.

## 4. Discussion

The concentrations of different milk fatty acids in our study (Table 1) seem slightly lower than those in other studies [5,28,29,30], which could be caused mainly by the differences in feed diet and milk-collection times of the farm, where they supplied their own total mix ration (TMR) three times per day, which is less than other similar Chinese Holstein cattle farms (four or five times per day). Compared to the results of Soyeurt et al. (2011) and Fleming et al. (2017), the variation coefficients ranged from 12.978% to 44.207% as fat-basis (g/100 g of fat), which were slightly lower, on average, than those in other studies. The higher variations of fatty acids as fat-basis (g/100 g of fat) in relation to those as milk-basis (g/100 g of milk) could be a tendency in which fatty acids exhibited high mean values and variation coefficients (Table 1).

Many previous studies have investigated the accuracy and applicability of prediction models based on R^2^ values. Soyeurt et al. (2011) suggested that models with R^2^ > 0.75 might be utilized for animal breeding. However, Zaalberg et al. (2021) used prediction models with R^2^ > 0.6 for mineral elements in animal breeding [31]. Cecchinato et al. (2009) showed low R^2^ values for curd characteristics predicted by MIRS, but they found high genetic correlations between the measured values and the predicted values [32]. In our study, 17 fatty acids (C8:0, C10:0, C12:0, C14:0, C18:0, C20:0, C22:0, C24:0, C18:1n9c, C20:1, C20:5n3, SFA, UFA, MUFA, SCFA, MCFA, and LCFA) showed RPD ≥ 2 and R^2^ ≥ 0.75 (Table 2), which is consistent with the results of Soyeurt et al. (2006). This suggests that these 17 fatty acids can be accurately predicted using MIRS, and that this method has the potential for further fat trait selections in animal breeding. Furthermore, 6 fatty acids (C12:0, C20:0, C22:0, C20:5n3, UFA, and LCFA) with R^2^ > 0.8, which were well predicted by MIRS, could also be used for breeding selections. For the grouped fatty acids, the R^2^ values of the test set were greater than 0.7 (Table 3), which is consistent with the results of Soyeurt et al. (2006), Rutten et al. (2009), and Fleming et al. (2017). For both the training and the test sets, 6 individual fatty acids (C20:0, C22:0, C24:0, C20:1, C18:3n6, and C20:5n3) as fat-basis (g/100 g of fat) showed R^2^ values greater than 0.7, whereas inconsistent results were found in other studies [4,5]. Fleming et al. (2017) obtained higher accuracy (R^2^) from fatty acids expressed on the milk-basis than on the fat-basis. Soyeurt et al. (2011) used the fatty acids predicted in milk for their prediction in fat and only achieved results better than those of the direct prediction in fat for C6:0, C12:0, C18:2 cis-9, cis-12, SFA, and SCFA. RPD is also used to measure the prediction effect and accuracy of models [33,34]. Three classifications of RPD are as follows: high prediction accuracy, which can be used for the quantitative prediction of substances when RPD ≥ 2; good prediction, which can be used for rough quantitative prediction or qualitative analysis when 1.4 ≤ RPD < 2; and low prediction accuracy, which cannot be used for quantitative prediction when RPD < 1.4. Generally, a higher accuracy (R^2^ and RPD) can also be observed in the prediction of fatty acids by MIRS on the milk-basis (*n* = 22) than on the fat-basis (*n* = 9) (Table 2 and Table 3), which is consistent with the results of other studies [4,5,21,35].

Different spectral pre-processing algorithms influence the prediction accuracy of fatty acids. Soyeurt et al. (2012) used MIRS to predict the lactoferrin content in bovine milk and obtained the highest prediction accuracy using PLSR based on DER1. Our study also found that derivatives (DER1 and DER2) and SG smoothing algorithms can be applied for most fatty acid predictions (Table 2). The derivative algorithm uses the absorbance values corresponding to each of two adjacent wave points to calculate their derivative values, where the spectrum is processed by the derivative. The wave points with large differences in absorbance reduce signal/noise interference; then, the corresponding value of the current wave point moves sequentially to retain the spectral information for stronger spectrum continuity (Figure 1).

## 5. Conclusions

In this study, different regression models led to varying prediction accuracy of fatty acid contents, while different pre-processing algorithms for the spectra also influenced prediction accuracy. It was revealed that a higher accuracy for most fatty acids can be achieved when derivative and SG pre-processing algorithms for RFR models were used. Therefore, after a series of evaluations in Chinese Holstein cows, these results suggest that the application of MIRS to predict the fatty acid contents of milk is feasible.

## Figures and Tables

**Figure 1 molecules-28-00666-f001:**
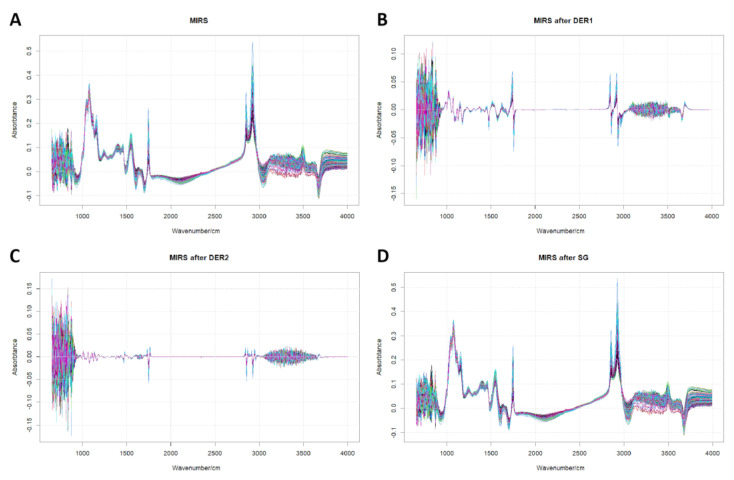
MIRS after DER1, DER2, and SG pre-processing algorithms. Note: MIRS, DER1, DER2, and SG indicate mid-infrared spectrum, first-order derivative, second-order derivative, and Savitzky–Golsy convolution smoothing, respectively.

**Figure 2 molecules-28-00666-f002:**
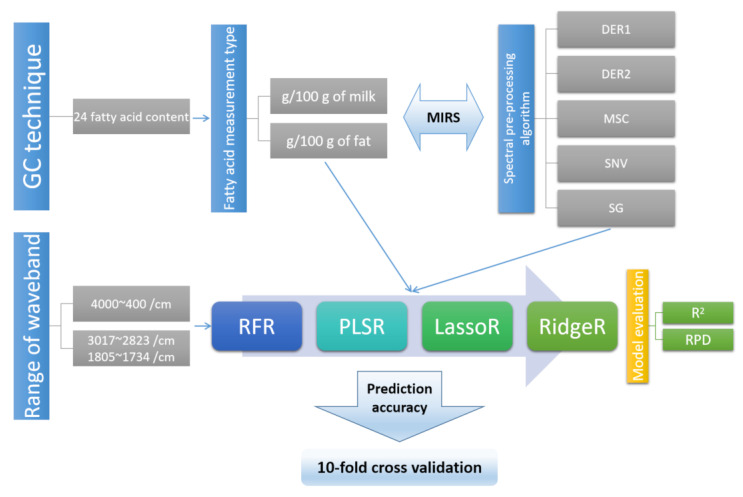
Summary of fatty acid prediction methods. Note: GC, MIRS, DER1, DER2, MSC, SNV, SG, RFR, PLSR, LassoR, RidgeR, R^2^, and RPD indicate gas chromatography, mid-infrared spectrum, first-order derivative, second-order derivative, multiple scattering correction, standard normal transform, Savitzky–Golsy convolution smoothing, random forest regression, partial least square regression, least absolute shrinkage and selection operator regression, ridge regression, determination coefficient, and residual predictive deviation, respectively.

**Table 1 molecules-28-00666-t001:** The minimum, mean, maximum, and variation coefficient of fatty acid contents measured by the GC technique.

Fatty Acid	Milk-Basis (g/100 g of Milk)					Fat-Basis (g/100 g of Fat)				
	Sample Size	Minimum	Mean	Maximum	Variation Coefficient (%)	Sample Size	Minimum	Mean	Maximum	Variation Coefficient (%)
C8:0	325	0.007	0.016	0.028	28.784	324	0.327	0.532	0.757	14.778
C10:0	323	0.013	0.044	0.082	35.416	326	0.598	1.402	2.324	21.273
C11:0	317	0.002	0.003	0.006	25.439	319	0.053	0.110	0.175	23.125
C12:0	321	0.019	0.062	0.115	34.871	327	0.829	2.018	3.323	21.721
C13:0	321	0.003	0.005	0.008	24.468	323	0.082	0.166	0.255	22.062
C14:0	322	0.094	0.231	0.371	27.975	321	4.058	7.546	11.358	15.075
C15:0	320	0.012	0.029	0.052	29.138	321	0.456	0.968	1.519	21.227
C16:0	325	0.366	0.877	1.491	28.154	323	17.710	28.620	42.251	13.802
C17:0	324	0.008	0.016	0.027	26.180	322	0.285	0.528	0.797	19.289
C18:0	320	0.098	0.313	0.600	35.024	326	4.070	10.254	17.150	25.195
C20:0	321	0.006	0.008	0.011	15.098	321	0.157	0.270	0.398	19.278
C22:0	332	0.004	0.005	0.006	9.413	329	0.074	0.172	0.286	25.529
C24:0	324	0.004	0.005	0.005	5.837	323	0.069	0.154	0.246	24.812
C14:1	325	0.006	0.019	0.033	32.157	317	0.230	0.610	1.138	31.033
C16:1	322	0.016	0.038	0.068	30.326	320	0.602	1.258	2.202	26.094
C18:1n9c	321	0.189	0.460	0.815	28.657	323	7.953	15.282	23.746	20.201
C20:1	321	0.003	0.003	0.005	13.134	319	0.065	0.116	0.190	24.333
C22:1n9	322	0.007	0.015	0.028	32.090	317	0.193	0.510	1.140	44.207
C20:3n6	322	0.003	0.006	0.009	24.212	322	0.109	0.192	0.288	18.233
C20:4n6	323	0.004	0.007	0.010	20.368	317	0.122	0.222	0.329	18.416
C20:5n3	332	0.002	0.003	0.004	10.163	323	0.046	0.094	0.149	23.530
C18:2n6c	323	0.030	0.070	0.120	28.437	327	1.052	2.300	3.654	18.149
C18:3n6	321	0.003	0.003	0.004	7.621	318	0.047	0.100	0.169	24.898
C18:3n3	324	0.004	0.008	0.013	24.352	324	0.155	0.278	0.417	16.480
SFA	325	0.692	1.627	2.714	28.415	322	29.494	52.710	74.351	13.287
UFA	323	0.288	0.638	1.090	26.277	326	13.162	21.266	31.904	18.082
MUFA	322	0.240	0.539	0.938	26.878	324	9.501	17.920	27.111	19.321
PUFA	324	0.046	0.098	0.159	25.514	325	1.720	3.196	4.863	16.229
SCFA	323	0.020	0.060	0.109	33.392	324	0.926	1.934	2.936	18.762
MCFA	325	0.580	1.269	2.147	27.475	322	25.617	41.372	61.272	12.978
LCFA	321	0.371	0.925	1.610	28.353	326	17.003	30.776	46.784	19.365

Note: SFA, UFA, MUFA, PUFA, SCFA, MCFA, LCFA, and GC indicate saturated fatty acid, unsaturated fatty acid, monounsaturated fatty acid, polyunsaturated fatty acid, short chain (4 to 10 carbons) fatty acid, medium chain (11 to 16 carbons) fatty acid, long chain (more than 16 carbons) fatty acid, and gas chromatography, respectively. The variation coefficient (%) is the ratio of standard deviation to the mean, which can be used to compare the degree of dispersion among the fatty acids.

**Table 2 molecules-28-00666-t002:** Best prediction accuracy for the optimal strategy in the test set for each fatty acid.

Fatty Acid	Pre-Processing Algorithm	MIRS Range (cm^−1^)	Model	Basis (g/100 g)	Test Set	
					R^2^	RPD
C8:0	SNV	3017~2823/1805~1734	PLSR	Milk	0.77	2.11
C10:0	DER1	3017~2823/1805~1734	RFR	Milk	0.77	2.07
C11:0	DER1	3017~2823/1805~1734	LassoR	Fat	0.55	1.48
C12:0	DER1	3017~2823/1805~1734	LassoR	Milk	0.84	2.50
C13:0	SG	3017~2823/1805~1734	PLSR	Milk	0.66	1.72
C14:0	DER1	4000~400	RFR	Milk	0.78	2.05
C15:0	SG	3017~2823/1805~1734	PLSR	Milk	0.57	1.53
C16:0	SG	3017~2823/1805~1734	RFR	Milk	0.75	1.98
C17:0	SG	3017~2823/1805~1734	LassoR	Milk	0.73	1.89
C18:0	DER1	4000~400	PLSR	Milk	0.77	2.08
C20:0	SNV	3017~2823/1805~1734	PLSR	Fat	0.82	2.35
C22:0	DER2	4000~400	RFR	Fat	0.86	2.66
C24:0	SG	4000~400	RFR	Fat	0.80	2.20
C14:1	MSC	3017~2823/1805~1734	PLSR	Fat	0.62	1.63
C16:1	SNV	3017~2823/1805~1734	LassoR	Milk	0.62	1.64
C18:1n9c	SG	3017~2823/1805~1734	LassoR	Milk	0.77	2.00
C20:1	DER2	4000~400	RFR	Fat	0.76	2.04
C22:1n9	DER1	4000~400	RFR	Fat	0.65	1.67
C18:2n6c	MSC	4000~400	RFR	Milk	0.63	1.61
C18:3n3	SG	4000~400	RFR	Milk	0.70	1.82
C18:3n6	DER2	3017~2823/1805~1734	RFR	Fat	0.76	2.00
C20:3n6	DER1	4000~400	RFR	Milk	0.62	1.61
C20:4n6	SNV	4000~400	RFR	Milk	0.50	1.42
C20:5n3	DER1	4000~400	RFR	Fat	0.91	3.06
SFA	SG	3017~2823/1805~1734	RFR	Milk	0.76	2.01
UFA	DER2	3017~2823/1805~1734	LassoR	Milk	0.82	2.15
MUFA	DER2	3017~2823/1805~1734	LassoR	Milk	0.79	2.06
PUFA	DER2	4000~400	RidgeR	Milk	0.71	1.75
SCFA	DER2	4000~400	RFR	Milk	0.77	2.04
MCFA	DER2	3017~2823/1805~1734	RFR	Milk	0.75	2.00
LCFA	DER2	3017~2823/1805~1734	RidgeR	Milk	0.83	2.29

Note: MIRS, DER1, DER2, MSC, SNV, SG, RFR, PLSR, LassoR, RidgeR, R^2^, and RPD indicate mid-infrared spectrum, first-order derivative, second-order derivative, multiple scattering correction, standard normal transform, Savitzky–Golsy convolution smoothing, random forest regression, partial least square regression, least absolute shrinkage and selection operator regression, ridge regression, determination coefficient, and residual predictive deviation, respectively.

**Table 3 molecules-28-00666-t003:** Best prediction accuracy of different prediction models for each fatty acid expressed as g/100 g of fat and g/100 g of milk, using training and test sets.

Fatty Acid	Pre-Processing Algorithm		MIRS Range (cm^−1^)		Model		Training Set				Test Set			
							R^2^		RPD		R^2^		RPD	
	Milk	Fat	Milk	Fat	Milk	Fat	Milk	Fat	Milk	Fat	Milk	Fat	Milk	Fat
C8:0	SNV	MSC	3017~2823/1805~1734	3017~2823/1805~1734	PLSR	LassoR	0.75	0.43	2.01	1.33	0.77	0.43	2.11	1.32
C10:0	DER1	DER1	3017~2823/1805~1734	3017~2823/1805~1734	RFR	LassoR	0.61	0.49	1.60	1.40	0.77	0.44	2.07	1.33
C11:0	DER2	DER1	3017~2823/1805~1734	3017~2823/1805~1734	LassoR	LassoR	0.57	0.51	1.53	1.43	0.53	0.55	1.46	1.48
C12:0	DER1	SNV	3017~2823/1805~1734	3017~2823/1805~1734	LassoR	LassoR	0.79	0.55	2.18	1.49	0.84	0.27	2.50	1.17
C13:0	SG	SNV	3017~2823/1805~1734	3017~2823/1805~1734	PLSR	LassoR	0.24	0.56	1.16	1.50	0.66	0.42	1.72	1.30
C14:0	DER1	DER1	4000~400	3017~2823/1805~1734	RFR	PLSR	0.66	0.16	1.72	1.10	0.78	0.43	2.05	1.34
C15:0	SG	MSC	3017~2823/1805~1734	3017~2823/1805~1734	PLSR	PLSR	0.45	0.25	1.37	1.17	0.57	0.32	1.53	1.22
C16:0	SG	DER2	3017~2823/1805~1734	4000~400	RFR	RidgeR	0.64	0.55	1.66	1.33	0.75	0.22	1.98	1.12
C17:0	SG	MSC	3017~2823/1805~1734	3017~2823/1805~1734	LassoR	PLSR	0.65	0.40	1.70	1.32	0.73	0.59	1.89	1.56
C18:0	DER1	SNV	4000~400	3017~2823/1805~1734	PLSR	LassoR	0.66	0.60	1.72	1.58	0.77	0.55	2.08	1.49
C20:0	SG	SNV	3017~2823/1805~1734	3017~2823/1805~1734	PLSR	PLSR	0.52	0.76	1.46	2.04	0.71	0.82	1.88	2.35
C22:0	DER2	DER2	4000~400	4000~400	RidgeR	RFR	0.70	0.83	1.76	2.42	0.52	0.86	1.44	2.66
C24:0	DER2	SG	4000~400	4000~400	RidgeR	RFR	0.64	0.90	1.55	3.20	0.61	0.80	1.46	2.20
C14:1	SNV	MSC	3017~2823/1805~1734	3017~2823/1805~1734	LassoR	PLSR	0.63	0.38	1.65	1.28	0.51	0.62	1.40	1.63
C16:1	SNV	MSC	3017~2823/1805~1734	3017~2823/1805~1734	LassoR	LassoR	0.54	0.38	1.47	1.27	0.62	0.55	1.64	1.50
C18:1n9c	SG	MSC	3017~2823/1805~1734	3017~2823/1805~1734	LassoR	LassoR	0.60	0.52	1.58	1.45	0.77	0.34	2.00	1.20
C20:1	SG	DER2	3017~2823/1805~1734	4000~400	PLSR	RFR	0.54	0.77	1.48	2.06	0.49	0.76	1.41	2.04
C22:1n9	DER2	DER1	4000~400	4000~400	RFR	RFR	0.51	0.53	1.43	1.45	0.45	0.65	1.36	1.67
C18:2n6c	MSC	SG	4000~400	4000~400	RFR	RidgeR	0.59	0.13	1.56	1.07	0.63	0.15	1.61	1.08
C18:3n3	SG	DER1	4000~400	3017~2823/1805~1734	RFR	RFR	0.60	0.17	1.59	1.09	0.70	0.27	1.82	1.13
C18:3n6	SG	DER2	3017~2823/1805~1734	3017~2823/1805~1734	RFR	RFR	0.18	0.84	1.08	2.47	0.14	0.76	1.04	2.00
C20:3n6	DER1	MSC	4000~400	3017~2823/1805~1734	RFR	PLSR	0.50	0.23	1.42	1.15	0.62	0.39	1.61	1.29
C20:4n6	SNV	SNV	4000~400	4000~400	RFR	PLSR	0.44	0.29	1.34	1.19	0.50	0.46	1.42	1.37
C20:5n3	DER2	DER1	4000~400	4000~400	RFR	RFR	0.33	0.83	1.23	2.41	0.43	0.91	1.29	3.06
LCFA	DER2	DER1	3017~2823/1805~1734	4000~400	RidgeR	RFR	0.68	0.41	1.78	1.31	0.83	0.42	2.29	1.32
MCFA	DER2	SNV	3017~2823/1805~1734	3017~2823/1805~1734	RFR	LassoR	0.64	0.23	1.67	1.14	0.75	0.28	2.00	1.18
MUFA	DER2	DER1	3017~2823/1805~1734	3017~2823/1805~1734	LassoR	LassoR	0.61	0.56	1.59	1.51	0.79	0.43	2.06	1.30
PUFA	DER2	SG	4000~400	3017~2823/1805~1734	RidgeR	RFR	0.71	0.16	1.80	1.08	0.71	0.16	1.75	1.07
SCFA	DER2	MSC	4000~400	3017~2823/1805~1734	RFR	LassoR	0.66	0.51	1.71	1.43	0.77	0.48	2.04	1.37
SFA	SG	SG	3017~2823/1805~1734	3017~2823/1805~1734	RFR	LassoR	0.66	0.32	1.73	1.21	0.76	0.25	2.01	1.16
UFA	DER2	MSC	3017~2823/1805~1734	3017~2823/1805~1734	LassoR	LassoR	0.62	0.42	1.62	1.31	0.82	0.48	2.15	1.38

Note: MIRS, DER1, DER2, MSC, SNV, SG, RFR, PLSR, LassoR, RidgeR, R^2^, and RPD indicate mid-infrared spectrum, first-order derivative, second-order derivative, multiple scattering correction, standard normal transform, Savitzky–Golsy convolution smoothing, random forest regression, partial least square regression, least absolute shrinkage and selection operator regression, ridge regression, determination coefficient, and residual predictive deviation, respectively.

**Table 4 molecules-28-00666-t004:** Seven classified fatty acid groups, according to hydrocarbon chain saturation and carbon chain length.

Fatty Acid Group According to Hydrocarbon Chain Saturation		Fatty Acid Group According to Carbon Chain Length	
SFA	C8:0, C10:0, C11:0, C12:0, C13:0, C14:0, C15:0, C16:0, C17:0, C18:0, C20:0, C22:0, C24:0	SCFA	C8:0, C10:0
UFA	C14:1, C16:1, C18:1n9c, C18:2n6c, C18:3n6, C18:3n3, C20:1, C20:3n6, C20:4n6, C22:1n9, C20:5n3	MCFA	C11:0, C12:0, C13:0, C14:0, C15:0, C16:0, C16:1
MUFA	C14:1, C16:1, C18:1n9c, C20:1, C22:1n9	LCFA	C17:0, C18:0, C18:1n9c, C18:2n6c, C20:0, C18:3n6, C18:3n3, C20:1, C22:0, C20:3n6, C20:4n6, C22:1n9, C20:5n3, C24:0
PUFA	C18:2n6t, C18:2n6c, C18:3n6, C18:3n3, C20:3n6, C20:4n6, C22:2, C20:5n3		

Note: SFA, UFA, MUFA, PUFA, SCFA, MCFA, and LCFA indicate saturated fatty acid, unsaturated fatty acid, monounsaturated fatty acid, polyunsaturated fatty acid, short chain (4 to 10 carbons) fatty acid, medium chain (11 to 16 carbons) fatty acid, and long chain (more than 16 carbons) fatty acid, respectively.

## Data Availability

Not applicable.

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
