# Peer review of "Predictions of Milk Fatty Acid Contents by Mid-Infrared Spectroscopy in Chinese Holstein Cows"

_molecules, 2023, doi:10.3390/molecules28020666_

Round 1
Reviewer 1 Report
Dear Author,
This work has lacks novelty. So many publications are already available on estimating fatty acid content in various types of milks including cows (Chinese Holstein cows; https://doi.org/10.3390/ani10010139) using mid-infrared spectrometry.
In this manuscript authors had tried to estimate 24 kinds of milk fatty acid concentrations by gas chromatography, which simultaneously produced MIRS from milk samples. However, the relevance of this study is questionable.
It is not clear that why authors have used only Chinese Holstein cow's milk, this makes it a regional study. Further what authors did new after the similar study published in, Animals 2020, 10, 139. https://doi.org/10.3390/ani10010139 and why authors have not cited this reference?
It is suggested that the authors should use a variety of milk e.g. Different kinds of cow milk, dairy milk, packed milk, Buffalow milk etc. which will make this study globally important.
Authors should include the latest publication on milk fatty acid content + mid-infrared spectroscopy.
With Regards,
Reviewer 2 Report
General comments
The paper covers a relevant area concerning the use of mid infrared spectroscopy to predict fatty acid concentrations. While such data has been widely reported over the last decade this would appear to be the first time it has been applied to Chinese cows.
Specific comments
Line 85: Suggest give information about what temperature the milk was stored at and how long after collection the respective analyses were performed
Line 88: Please give reference to the GC method (eg ISO 15885: 2002/IDF 184: 2002)
Line 97: Please explain more clearly why there was two groups of mid infrared spectral data (is it wavebands as indicated in line 114?)
Line 106: Please give further details on how fat was determined by MIRS (eg instrument and calibration) and clarify why there and what were missing values.
Line 110: Were any of the preprocessing procedures used in conjunction with each other as opposed to individually, if not why not?
Table 2: It can be useful to give the range (minimum and maximum) and mean of fatty acid results
Line 154: Greater accuracy when predicting fatty acids by MIR expressed on a milk basis has ample precedence in the literature and should be referenced (the Soyeurt 2011 reference already cited)
Table 3 and associated text: R2 and RPD values that different at the three significant figures level are probably not significantly different, this should be stated in text and suggest only quote to two significant figures.
Line 203: Rather than error its possible that lower fat is due to diet and location of the cows
Line 227: Can you suggest why these six studies were inconsistent? Could again be due to variation in diet between European and Chinese cows
Line 237: “The wave points with large differences in absorbance reduce noise interference,” not true rather the signal/noise ratio is better in these locations
Reviewer 3 Report
The overall novelty of the manuscript is appropriate. In conclusion doesn't use words like we or ours.
Round 2
Reviewer 1 Report
NA